# One Earth–One Health to Combat Antimicrobial Resistance Having a Dual Mutation Pattern, Based on the Robust Self-Regulation and Perpetual Reconstruction of Ecosystems

**DOI:** 10.3390/antibiotics14090920

**Published:** 2025-09-11

**Authors:** Ganjun Yuan, Jianing Hu, Meifang Zhang, Xingguyue Chen, Yumei Luo

**Affiliations:** 1Biotechnological Engineering Center for Pharmaceutical Research and Development, Jiangxi Agricultural University, Nanchang 330045, China; hjian@stu.jxau.edu.cn (J.H.); 0202023095@stu.jxau.edu.cn (M.Z.); starry.moon@stu.jxau.edu.cn (X.C.); 2Laboratory of Natural Medicine and Microbiological Drug, College of Bioscience and Bioengineering, Jiangxi Agricultural University, Nanchang 330045, China; luoym135@163.com

**Keywords:** OE-OH, one health, antibiotic resistance, microbe, plant, Lamarck, niche, Darwin, environment, combination

## Abstract

**Background/Objectives:** Antimicrobial resistance (AMR) has emerged as a grave threat to human health, and a One Earth–One Health (OE-OH) concept was proposed for addressing this challenge in 2024. Here, this concept was systematically defined, clarified, and refined, for better understanding, interpreting related results, and taking some measures to combat the crisis. **Methods**: Using logical reasoning and deductive methods, a dual mutation pattern was put forward for microbial resistance, adhering to the principle of parsimony and integrating Lamarckian, Darwinian, and Niche construction theories, and the evolutionary origins of current AMR were schematically presented. Subsequently, its theoretical foundation, together with a fundamental mathematical model, was defined and clarified based on the robust self-regulation and perpetual reconstruction of ecosystems, and then the generation, dissemination, and elimination of AMR and antibiotic resistance genes (ARGs) were sorted out and elucidated from abiotic and biotic factors. Finally, learning from the approach of problem management, some crucial measures are suggested for the research and development, application, and management of antibiotics, emphasizing the key role of simulating and utilizing the self-regulation of ecosystems. **Results**: A dual mutation pattern of microbial resistance and the evolutionary origins of current AMR was put forward. The theoretical foundation of the OE-OH concept, together with a fundamental mathematical model, was presented. Some unique perspectives, such as the emergence of AMR and ARGs 3.5 billion years ago and their ubiquity across the globe prior to antibiotic use, were clarified. Moreover, some crucial measures are proposed for addressing AMR. **Conclusions**: It is essential to implement the OH Joint Plan of Action from the OE-OH perspective, strongly emphasizing the key role of simulating and utilizing the self-regulation of ecosystems on addressing AMR.

## 1. Introduction

Antimicrobial resistance (AMR) has emerged as a serious threat to global public health and economic development, and the COVID-19 pandemic has further exacerbated this crisis [1,2,3]. It is shown that AMR has rapidly spread from diverse settings where antibiotics are used into surrounding environments [4,5]. Moreover, antibiotic resistance genes (ARGs) can be extensively detected in a wide range of water and soil environments [4,5,6,7], with their presence even being on the north slope of Mount Everest [8]. In light of these alarming trends, the World Health Organization (WHO) has projected that, without any intervention, antibiotic resistance would lead to 10 million deaths annually by 2050 [9,10]. In response to this crisis, the Food and Agriculture Organization of the United Nations (FAO), the United Nations Environment Programme (UNEP), the WHO, and the World Organisation for Animal Health (WOAH, founded as OIE) have developed the One Health Joint Plan of Action (2022–2026) (OH JPA) and appeal working together for the health of humans, animals, plants, and the environment [7,11]. The OH JPA outlines the commitment of the four organizations to collectively advocate and support the implementation of One Health (OH).

As is widely acknowledged, the use of antibiotics, particularly their overuse and abuse, has caused the crisis of AMR. However, it is worth exploring whether these reports can accurately reflect the actual situation of AMR. On the one hand, many studies, shaped by the interests of research, focus on the spread and evolution of AMR among microorganisms and in the environment, understudying the self-regulatory capacity of the Earth’s ecosystem in weakening and eliminating AMR and ARGs [12,13,14,15]. This self-regulatory capacity encompasses the complex interactions among humans, plants, animals, microorganisms, and the environment, as well as the functional redundancy and removal of ARGs [4,16,17,18]. On the other hand, it is also worth contemplating whether the mere detection of ARGs implies that AMR has been transmitted to the sampling site. Both aspects have prompted a more profound re-evaluation for the strategies and measures aimed at addressing AMR. Based on the discoveries of various laws on drug combinations preventing AMR and the effects of numerous plant metabolites on reversing AMR [12,19,20,21,22], a One Earth–One Health (OE-OH) concept was put forward for preventing AMR, without details, during the 6th International Caparica Conference in Antibiotic Resistance 2024 (IC^2^AR 2024) and in a following publication [12,23].

OE-OH is a systems engineering approach and concept that takes niche construction as its core theoretical foundation. It progresses from exploring, understanding, and interpreting the self-regulation and continuous reconstruction mechanisms of ecosystems to simulating and applying these mechanisms, ultimately aiming to promote human health and sustainable development, particularly in addressing the crisis posed by AMR [12,23]. It adheres to holism and systems thinking, regarding the Earth as a giant ecosystem composed of countless sub-ecosystems that operate through analogous mechanisms and involve diverse abiotic and biotic factors such as soil, climate, air, sunlight, water, plate tectonics, geological disasters, microbes, plants, animals, and/or humans. It believes and emphasizes the strong regulatory capacity of the Earth’s ecosystem and advocates the adoption of eco-mimetic methods and strategies in addressing health threats.

The OH concept posits that AMR encompasses multiple facets, including humans, animals, plants, and the environment, emphasizing the pivotal role of humans in combating AMR. In contrast, OE-OH places greater emphasis on the robust self-regulatory capacity of the Earth’s ecosystem and its sub-ecosystems in relation to AMR. Specifically, it accords equal importance to the generation, spread, weakening, and elimination of AMR within the ecosystem, and regards the production and use of antibiotics as an intervention on the ecosystem’s AMR dynamics. Moreover, in approaches and strategies for addressing AMR, OE-OH attaches great importance to simulating and applying the regulatory mechanisms of ecosystems. Therefore, OE-OH can serve as an important reference in approach and concept for ensuring the scientific, rational, and sustainable implementation of the OH JPA. Herein, the theoretical foundation of this concept, together with many innovative prospectives, reasoning, deductions, and inductions concerning AMR and ARGs, are presented, from the OE-OH concept, as follows. Based on these, some distinctive strategies for combatting AMR are subsequently proposed and elaborated.

## 2. The OE-OH Concept

### 2.1. A Dual Mutation Pattern of Microbial Resistance

In accordance with the principle of parsimony [24,25], also known as Occam’s Razor, the mutation theory of bacterial resistance can be reconsidered. We believe that microbial resistance predominantly engages in proactive evolution by adaptive mechanisms, which are designed to avert the unnecessary expenditure of energy and resources that are typically associated with random mutations. This assertion aligns with Lamarck’s theory, which has also garnered support from recent research endeavors [26]. However, there may also be minor imprecise mutants during the course of proactive resistance mutations. Simultaneously, some ARG-carrying mutants can also emerge through the occasional and non-adaptive random mutations and be passively screened by natural selection in accordance with Darwinian evolution. From these, ARG-carrying mutants originate from two distinct types of mutations, as shown in Figure 1A. The dual mutation pattern of microbial resistance, characterized primarily by proactive evolution and occasionally by passive selection, exhibits a striking congruence with the niche construction theory, which pertains to ecosystems [27,28]. From this pattern, the evolutionary origins of current AMR were deduced and is shown in Figure 1B. Moreover, although proactive mutation is the primary pattern of microbial resistance to antibiotics, a large number of antibiotic-resistant microbes can still be found prior to exposure to antibiotics. This is due to the substantial accumulation of ARG-carrying pathogens in the Earth’s ecosystem over a long period of time.

### 2.2. Theoretical Logic of the OE-OH Concept Based on Ecosystems

Inspired by the dual mutation pattern illustrated in Figure 1, it can be inferred that antimicrobial metabolites produced by microbes (some of which are termed antibiotics now), microbial resistance, and ARGs likely emerged concurrently with the formation of microbial ecosystems. This is because they arise from the competition among microorganisms within ecosystems and are renewed as ecosystems undergo reconstruction [29,30,31,32]. Namely, ARG-carrying bacteria likely first emerged when certain microbial ecosystems were formed 3.5 billion years ago [33] and have been continuously spreading and renewing ever since (Figure 2). This was also supported by a recent publication [34]. Simultaneously, the Earth, including soil, various environments, and all the organisms that inhabit it, can be regarded as a giant ecosystem composed of countless sub-ecosystems that operate through similar mechanisms. Therefore, the robust self-regulatory capacity and perpetual reconstruction of ecosystems, particularly those of the Earth’s [13,14,15], form the theoretical logic of the OE-OH concept. From this perspective, various aspects related to microbial resistance can be reexamined.

### 2.3. A Fundamental Mathematical Model for the ARGs Renewing with the Ecosystem

The OE-OH concept envisions the Earth as a giant ecosystem, intricately woven from a myriad of sub-ecosystems that operate through similar mechanisms. The amount of ARGs within an ecosystem can be articulated through a fundamental mathematical model as follows.ARG_i_ = ARG_0_ + (ARG_1_^In^ + ARG_2_^In^ + …… + ARG_i_^In^ + …… + ARG_n_^In^) + (ARG_1_^De^ + ARG_2_^De^ + …… + ARG_i_^De^ + …… + ARG_n_^De^)$${\mathrm{ARG}}_{n}={\mathrm{ARG}}_{0}+\sum_{i=1}^{n}{ARG}_{i}^{In}+\sum_{i=1}^{n}{ARG}_{i}^{De}$$
where ARG_i_ and ARG_n_ are the amount of ARGs at two time points (Figure 2 and Figure 3) of the ecosystem, which can be the Earth’s ecosystem or its various sub-ones, and time point i can be equal to n; ARG_0_ can be the amount of ARGs at any time point, while time points i and n are more than or equal to time point 0, and especially, the amount of ARG_0_ is zero before the microbial ecosystem emerged on Earth; ARG^In^ is the increased amount of ARGs within the ecosystem, for example, ARG_1_^In^ is that from time point 0 to 1; ARG^De^ (defined as a negative value) is the decreased amount of ARGs within the ecosystem, for example, ARG_1_^De^ is that from time point 0 to 1; $\sum_{i=1}^{n}{ARG}_{i}^{In}$ is the sum of the increased amount of ARG_s_ from time point 1 to n, and $\sum_{i=1}^{n}{ARG}_{i}^{De}$ is the sum of the decreased amount of ARG_s_ from time point 1 to n.

## 3. ARG Analyses from the OE-OH Concept Based on Ecosystems

### 3.1. ARGs Emerging Prior to the Emergence of Humans and Existing Everywhere in the World

Antimicrobial metabolites are a category of natural products that generate from the competition for ecological niches among microorganisms within ecosystems or in response to survival stress [29,30,31]. As noted earlier, microbes emerged 3.5 billion years ago, and by that time, diverse ecosystems teeming with microbial communities had already taken shape [33,34]. After a long period of evolution, there are sufficient reasons to infer that the structural skeletons of most clinical antibiotics can be biosynthesized by environmental microorganisms. This can be confirmed by the fact that the structural skeletons of clinical antibiotics were mostly discovered from soil microorganisms [29]. In other words, microorganisms that carry genetic information for biosynthesizing the structural skeletons of most clinical antibiotics, including microbial strains belonging to the same or different genera and species, are widely distributed in various sub-ecosystems of Earth. Even synthetic quinolone antimicrobial agents bear a resemblance in structural framework to naturally occurring compounds such as plant-derived flavonoids and benzofuranones [12,35]. This is also a reason that natural products with antimicrobial activity can be continuously discovered from environmental microorganisms. Likely, the genetic information of microorganisms producing these natural products has existed on the Earth before the emergence of humans [36,37,38] but was only discovered and excavated after entering into the 20th century.

It is reported that today’s pathogenic microorganisms originate from the soil [39,40]. Therefore, clinical pathogens should be widely distributed across various ecosystems on Earth and likely prior to the emergence of humans [41]. Since antimicrobial metabolites were generated with the formation of microbial ecosystems, it is reasonable to deduce that, due to microbial competition within ecosystems, the ARGs of pathogenic microorganisms evolved in response to the survival stress from antimicrobial metabolites (some of which are now used as antibiotics) produced by other environmental microbes may have already emerged at that time.

As the Earth’s ecosystem and its diverse sub-ecosystems evolve and undergo reconstruction driven by a variety of biotic and abiotic factors [42], new ARGs are perpetually emerging. Moreover, ARGs are continuously spreading and renewing across every corner of the Earth. Therefore, it can be inferred that microorganisms responsible for producing the structural skeleton of certain antibiotics, along with pathogens carrying corresponding ARGs, are located in specific ecosystems. When antibiotics are isolated from soil or marine microbes, it is likely that these pathogens have already encountered these antibiotics during the evolutionary process of the Earth’s ecosystem. Through proactive adaptive mutations and the acquisition of heritable ARGs, these pathogens have also developed resistance to these antibiotics. This may explain why ARG-carrying pathogens can sometimes be detected shortly after the introduction of new antibiotics to the market [43], or before they appear to have come into contact with the corresponding antibiotics, sometimes even before the antibiotics are officially approved [44]. Namely, these pathogens have, in fact, been exposed to the corresponding antibiotics long ago.

Given this, it is foreseeable that ARGs can be found not only in extreme environments such as Mount Everest and the Mariana Trench but also in the Antarctic and Arctic regions of the Earth. This notion is also supported by recent research [45,46,47,48,49] and means that ARGs are likely to be detected in varying degrees from every ecological niche containing microbes if the detection methods are sufficiently sensitive. Moreover, it is reasonable to assume that a certain amount of ARGs existed in various ecosystems and across the Earth prior to the production and use of antibiotics by humans (Figure 3).

### 3.2. ARGs by the Self-Regulation of Ecosystems Before the Use of Antibiotics

Since the emergence of microbes on Earth, diverse ecosystems composed of microbial communities have gradually taken shape and continuously evolved along with the incorporation of new organisms and the elimination of existing ones. Throughout the Earth’s evolutionary history, information on antibiotic resistance genes (ARGs) have been in a state of perpetual renewal, driven by the ongoing reconstruction of ecosystems. This renewal process encompasses the emergence and dissemination of new ARGs, as well as the weakening and elimination of existing ones, which is also supported by recent research [4,6,50,51,52,53,54].

As shown in Box 1, a wide range of biotic and abiotic factors, including the abiotic environment, water bodies, climate, microbes, plants, animals, and humans, can impact the renewal of ARGs within the Earth’s ecosystem and its various sub-ecosystems. Owing to the sufficient self-regulation, self-balancing, and buffering capacities of the Earth’s ecosystem [12,13,14,15], the evolution of AMR and ARGs has historically maintained a balanced and controllable state prior to the industrial production and use of antibiotics by humans. However, the situation has become increasingly worrying due to the extensive use of antibiotics, particularly their overuse and abuse in clinical settings, livestock, poultry farming, and aquaculture. This was also reflected in the rapid increase in AMR and the widespread dissemination of ARGs after the industrial production and use of antibiotics.

Box 1The OE-OH concept based on the self-regulation of the Earth’s ecosystem for AMR and ARGs.
ARG_n_ = ARG_0_ + $\sum_{i=1}^{n}{ARG}_{i}^{In}$ + $\sum_{i=1}^{n}{ARG}_{i}^{De}$Factors for the generation and spread of ARGs (ARG^In^) Factors for the weakening and elimination of ARGs (ARG^De^)
**Abiotic factors**

**Abiotic factors**
**The abiotic environment and water bodies**    Plate tectonics and plate motion, geological disasters    Soil components, dust, and air [6,39,55,56,57,58,59]    Hydrographic system and water flows [5,60,61]    Ocean and its currents [62,63,64]    The melting of ice and snow in the Arctic, the Antarctic, and high mountains [8]**Climate**    Rainfall, wind, global warming, and drought [42,65]    Biotic factors influenced by climate change and climate extremes [66,67,68,69]**The abiotic environment and water bodies**    Plate tectonics and plate motion, geological disasters    Soil components, dust, and air [6,55,70]    Dilution and/or elimination of antibiotics and ARGs [71,72,73,74]     Dilution, decrease, and disappearance of selection pressure [51]    The killing and clearance of ARGs and antibiotic-resistant pathogens [72,75,76,77,78]**Climate**    Rainfall, wind, global warming, and drought [42,65,76]    Biotic factors influenced by climate change and climate extremes [51,67,68]

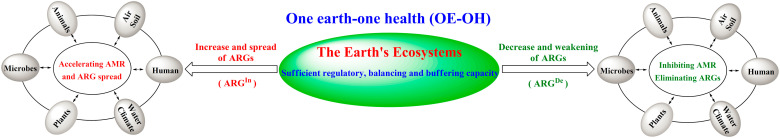


**Biotic factors**

**Biotic factors**
**Microbes**    Bacteriophage and virus [79,80,81]    The interaction of various microbes [29,30,31,67,82,83]    Amino acid auxotrophic microbes [84]    The activation to ARG expressions by antimicrobial metabolites [18]    Horizontal transfer of ARGs according to genetic compatibility and ecological connectivity [54]**Plants**    The decrease in plant diversity [85]    ARG enrichment in the rhizosphere of plants [86]    The activation of ARG expressions by plant antimicrobial metabolites [18]    The interactions between plants and microbes**Animals**
    Activities and migration     Foraging and excretion [87,88,89,90,91,92,93]    The activation of ARG expressions by antimicrobial peptides [18]**Human and human activity**    Daily activities, migration, travels, diet, and excretion [3,39,87,88,90,92,94,95,96]    The activation of ARG expressions by antimicrobial peptides [52]    Population gathering and domestic sewage [4,97,98]    The use and abuse of antimicrobial agents in medical activities, livestock, poultry farming, aquaculture, etc. [4,51,62]    The use of air and environmental disinfectants [99]    The pollution of air and water from the production and use of antibiotics and environmental disinfectants [3,4,71,99,100,101,102]    The impacts of human activities on all biotic and abiotic factors that generate and spread AMR and ARGs in the Earthʹs ecosystem [63,66,85,103,104,105]**Microbes**    Fitness costs [50]    Negative selection and disappearance of selection pressure [16,17,18,51]    Genetic incompatibility inhibiting the horizontal transfer of ARGs [54,106]    The degradation of interactions among various microbes with antibiotics [82,107]    The killing and inhibition of other microbes and pathogens within ecosystems [4]**Plants**    The increase in plant diversity [85]    The degradation of antimicrobial agents by plants [108,109,110]    Sensitization of antimicrobial resistance by plant antimicrobial metabolites [12,111]    The killing and clearance of plant antimicrobial metabolites to drug-resistant pathogens [4,112]**Animals**
    The killing and clearance of animal bodies to drug-resistant pathogens       including the immune system, antibacterial peptides, beneficial microorganisms, etc. [113]    Foraging and sensitization of the intestinal microbial ecology to drug-resistant pathogens [4]**Human and human activity**    The killing and clearance of human bodies to antibiotic-resistant pathogens [52,53,114,115]          including the immune system, antibacterial peptides, beneficial microorganisms, etc.    Diet and sensitization of gut microbial ecology to antibiotic-resistant pathogens [12,116]    Degradation, reduction, and removal of antibiotics in waste residues and water from medical activities, livestock, poultry farming, aquaculture, etc. [72,107,108]    Disinfection and inactivation of antibiotic-resistant pathogens    All other possible human activities on biotic and abiotic factors that clear antibiotics and decrease ARGs within the Earthʹs ecosystems [72,104,113,117]
Note: OE-OH, One Earth-One Health; AMR, antimicrobial resistance; ARGs, antibiotic resistance genes; ARG^In^, the increased amount of ARGs within the ecosystem; and ARG^De^, the decreased amount of ARGs within the ecosystem.


### 3.3. Impact of Antibiotic Use on the Self-Regulation of Ecosystems to ARGs

In the 20th century, many secondary metabolites with antimicrobial activities at low concentrations were discovered from environmental microorganisms, particularly actinomycetes, fungi, and bacteria, across diverse habitats such as land, oceans, and the body surfaces and feces of animals and humans [29,31]. Some of these metabolites and their derivatives have been developed as clinical antibiotics. In fact, the genetic information of microorganisms producing these antibiotics, together with the corresponding ARGs carried by pathogens resistant to these antibiotics, was already generated long ago and can be considered as the products of competition among various microorganisms during the Earth’s evolutionary history. Although most of the clinical antibiotics currently in use are structural derivatives of these natural antibiotics, they share similar structural skeletons with corresponding naturally sourced antibiotics. Consequently, the information of their corresponding ARGs carried by pathogens has existed in nature long ago. Moreover, pathogens carrying these resistance genes have spread globally, with some disseminating through soil and others colonizing specific parts of the human body [39].

From the above, the widespread use of antibiotics across various domains, including medical practices, livestock breeding, poultry farming, and aquaculture, can be considered as human intervention on the evolution of ARGs within the global ecosystem [82]. In addition, general human activities, as well as the interactions between the human body and microbes, have been driving the spread of ARGs (Box 1) prior to the use of antibiotics; several key activities associated with the industrial production and use of antibiotics by humans have led to a sharp increase in AMR and the rapid spread of ARGs. These activities are as follows: (1) the use, overuse and abuse of antibiotics in medical practices, livestock, poultry farming, and aquaculture, etc. [4,51,62]; (2) the pollution of the Earth’s environment caused by wastewater discharged from the settings of antibiotic production and use [3,4,7,62,71,100,101,102]; (3) the excessive density and size of urban populations, and the large amounts of domestic sewage generated under conditions of antibiotic overuse and misuse [4,97,98]; and (4) the impacts of other human activities on all biotic and abiotic factors that contribute to the generation and spread ARGs within the Earth′s ecosystem [63,66,85,103,104,105].

Theoretically, the wider the application scope of antibiotics and the greater their usage, the larger the intervention intensity on ecosystems. When the intervention intensity keeps within a specific and controllable range, the Earth’s ecosystem and its myriad of sub-ecosystems have sufficient self-regulation capabilities to restore or reestablish a new balance (Box 1). But if the intervention intensity exceeds the self-regulation capacity of ecosystems, the balance of these ecosystems will inevitably be disrupted, endangering larger ecosystems and even the entire Earth’s ecosystem [13,118]. Therefore, our endeavors to discover new antibiotics seem to have the potential to solve the problem of antibiotic shortage caused by AMR, however they are merely a passive defense strategy even if the development of new antibiotics can be accelerated with the assistance of artificial intelligence (AI) [119]. Although AI has the potential to predict microbial resistance to existing antibiotics and facilitate a proactive defense in the future, the struggle between humans and microbes is at most evenly matched. More significantly, the damage to various ecosystems caused by the overuse and abuse of antibiotics, together with the impact and deterioration of new ecosystem reconstruction on human living conditions, should arouse sufficient attention in this struggle.

Therefore, if the overuse and abuse of antibiotics are not controlled, the approval and application of new antibiotics will only accelerate the spread of AMR and ARGs. This will lead to the continuous destruction of larger and more ecosystems centered on the application environment. Moreover, various ecosystems containing more pathogens carrying ARGs will be restructured, ultimately posing a threat to the survival and development of humans. Thus, it is imperative to implement scientific and rational measures to control the spread of AMR and ARGs, to maintain the balance of the Earth’s ecosystem and its various sub-ecosystems.

## 4. Measures Combating AMR from the OE-OH Concept Based on Ecosystems

As previously stated, the use of antibiotics can be regarded as an intervention on ecosystems from the OE-OH concept. Thereby, a counteracting intervention aimed at preserving the balance of ecosystems should be taken for effectively combating AMR (Figure 4) [120]. Simultaneously, the self-regulation and -balance capabilities of ecosystems can be fully understood and utilized for risk evaluation on some measures taken for the research and development, application, and management of antibiotics. Learning from the approach of problem management [121,122], some crucial measures from the OE-OH concept, with the support of the literature, are suggested as follows:

### 4.1. Minimizing Antibiotic Use, While Fully Utilizing the Regulatory Role of Plants

From the OE-OH perspective, the bodies of humans or animals themselves can be regarded as ecosystems with sufficient self-regulation capabilities. For instance, their immune systems [52], antimicrobial peptides [114], colonized probiotics, and the competition among microorganisms within the body’s ecosystem can eliminate infected pathogens, including those that are antibiotic-resistant. Meanwhile, many plants, such as traditional Chinese medicines and ethnic medicines, also have the potential to regulate the balance between the bodies of humans or animals and microbes [12,112,123]. This can be achieved through their regulatory effects on the body’s inflammatory and immune responses, their direct antibacterial, antifungal, and antiviral actions, as well as their ability to enhance the inhibitory activities of metabolites produced by probiotics against antimicrobial-resistant microbes. This has been evidenced in China during the COVID-19 pandemic [124]. Therefore, it is entirely feasible to reduce the use of antibiotics through utilizing the self-regulation and self-balance abilities of both humans and animals themselves, as well as the regulatory effects of plants on these abilities. Furthermore, even in cases where the bodies of humans and animals are infected, the balance between them and microbes can still be regulated or restored by plants. This can help to mitigate the progression of microbial infections, to the greatest extent possible, and avoid the unnecessary use of antibiotics.

### 4.2. Minimizing Antibiotic Emissions, While Fully Utilizing the Self-Regulation of Ecosystems

From the OE-OH perspective, microbial resistance predominantly follows the proactive pattern of adaptive evolution, occasionally the passive one of random mutation by natural selection. As the Earth continues to evolve and develop, ARG-carrying pathogens that are widely distributed can be activated, screened, and enriched under the stress of antibiotics. Consequently, the emission of antibiotics into the surrounding environment not only stimulates the overexpression of ARGs in pathogens and enriches the information of ARGs but also enables susceptible bacteria to proactively evolve into resistant pathogens. Therefore, it is crucial to minimize the emission of antibiotics and ARGs.

Alternatively, from the OE-OH perspective, the transmission processes of antibiotics and ARGs also involve their dilution, redundancy, or disappearance by various ecosystems. Therefore, when effective control is challenging and emissions are unavoidable, the weakening and elimination capacities of ecosystems to antibiotics and ARGs can be fully utilized [123,125,126,127]. At this moment, scientific and rational measures should be adopted, based on risk management [128], to implement graded emissions for keeping the emissions of antibiotics and ARGs within the controllable and balanced range of ecosystems, minimizing the damage caused by the excessive accumulation of antibiotics and ARGs to the original ecosystem.

### 4.3. Avoiding Excessive Density and Size of Urban Population

From the OE-OH perspective, the entire Earth’s ecosystem possesses a robust self-regulation capacity to manage the generation, dissemination, enrichment, dilution, weakening, and elimination of ARGs, thereby maintaining its equilibrium. However, the overuse and abuse of antibiotics by humans will lead to the continuous enrichment and spread of ARGs and stimulate their proliferation in the living environment [129], resulting in an increasing diversity and abundance of ARGs around their habitat. If the density and size of urban populations become excessive at this moment, they will significantly surpass the capacity of various biotic and abiotic factors (Box 1) to weaken and eliminate ARGs, as well as the self-regulation ability of ecosystems, leading to an excessive accumulation of ARGs in ecosystems centered around the gathering areas of urban populations. Therefore, it is crucial to avoid the excessive density and size of urban populations [126]. Also, it is better to be kept within reasonable limits for the scale of cities, and the urban layouts should be appropriately decentralized. This can also be indirectly proved by the transmission patterns of the COVID-19 pandemic. Therefore, it may be encouraged to conduct research on the impact of urban size and distribution spacing, as well as the density, distribution, and size of urban populations, on the enrichment and dissemination of ARGs.

### 4.4. Accelerating the Antibiotic Reserve Based on the Understanding for Microbial Defense Mechanisms

From the OE-OH perspective, it is unrealistic for humans to discover antibiotics to which microorganisms will never develop resistance. This also holds true for multi-target antibiotics whether they are natural or fully synthetic products. Nevertheless, to combat AMR, it remains necessary to develop new antibiotics that are difficult to trigger a resistance to, as much as possible, and a sufficient reserve of these new antibiotics should be ensured. To achieve this, we can gain a comprehensive understanding of the proactive defense mechanisms of microbes in ecosystems [18] and thoroughly explore the unknown ARGs within ecosystems to avoid, as much as possible, using new antibiotics to treat infections caused by pathogens carrying the corresponding ARGs. To attain this, we can fully utilize AI technology to comprehensively predict potential ARGs of pathogens and possible AMR [130,131]. Moreover, it is also encouraged to develop new antibiotics with high specificity and minimal disruption to gut microbes and organismal ecosystems [132]. In addition, from a policy standpoint, the protection period of patents for new antibiotics can be extended, for reducing the unnecessary use of new antibiotics.

### 4.5. Encouraging Antibiotics Used in Combination with Plant-Derived Antimicrobial Ingredients

From the OE-OH perspective, the increased application of new antibiotics will accelerate the spread and enrichment of AMR and ARGs if the overuse and abuse of antibiotics are not controlled. Therefore, it is imperative to use antibiotics rationally. Among various strategies for the rational use of antibiotics, combination therapy has the advantages of cost-effectiveness in enhancing the efficacy of antibiotics, reversing microbial resistance, and extending the life cycle of antibiotics, buying more time for the development of new antibiotics. This can also be proved by the clinical practice of combination therapy, such as the combination of sulfamethoxazole and trimethoprim, β-lactamase inhibitors and β-lactam antibiotics, and multiple anti-tuberculosis drugs. Therefore, combination therapy is highly commendable [12,82]. However, it is noteworthy that an inappropriate antibiotic combination would instead increase the risk of AMR, due to the effect of preventing resistance being associated with the fractional inhibitory concentration index, as well as the proportion and concentration of two antibiotics in the combination [19,20,133].

Antibiotics are derived from the competition among microorganisms. Both bacteria/fungi producing the structural skeleton of antibiotics and pathogenic bacteria are classified as microbes. As a result, their individual defense mechanisms are familiar to each other, making it easier for pathogens to develop resistance to antibiotics in the combination. However, the antimicrobial ingredients of plants originate from the interaction between microorganisms and plants within ecosystems, and the defense mechanisms between plants and microbes are less familiar to each other. Simultaneously, plant-derived antimicrobial components generally exhibit weaker antibacterial activity and smaller stresses on microbial survival compared with antibiotics. Therefore, it is more challenging for microbes to develop a resistance to them [20]. Moreover, the combination of plant-derived antibacterial components and antibiotics often has a wide range of synergistic effects [12], and the impact on gut microbes is also milder. Therefore, plant-derived antibacterial components are ideal candidates for combination therapy with antibiotics.

### 4.6. Simulating the Elimination of Antibiotics and ARGs Within Ecosystems

From the OE-OH perspective, ecosystems have robust capacity to regulate AMR, encompassing the dilution, weakening, and elimination of ARGs. Therefore, it is highly encouraged to simulate the elimination of antibiotics and ARGs within ecosystems based on a thorough understanding of their self-regulation mechanisms [120,134]. For instance, employing microbial ecology to combat ARG dissemination [4], utilizing photocatalysis-enhanced constructed wetlands to remove ARGs [72], simulating sunlight-induced inactivation of tetracycline-resistant bacteria [75], and leveraging bacteria–microalgae–fungi symbionts or plants to remove antibiotics [107,108]. Additionally, these ecological simulation methods and technologies can also, to the greatest extent possible, avoid the potential adverse effects on ecosystems that may be caused by the measures taken.

## 5. Methods

### 5.1. The Definition, Refinement and Clarification of the OE-OH Concept

The OE-OH concept was put forward from the previous results and the sufficient self-regulation ability of the Earth’s ecosystem in the evolution of AMR [12,23]. Here this concept has been further defined, improved, and clarified using logical reasoning and deductive methods based on the principle of parsimony [24,25], involving the understanding integration of Lamarck’s theory, Darwinian evolution, and the niche construction theory. It includes the dual mutation patterns of microbial resistance, the theoretical underpinnings of the OE-OH concept based on ecosystems, and a basic mathematical model for the ARGs renewing with the ecosystem. The literature supporting this reasoning and deduction, together with all other literature, was unsystematically searched from PubMed database and Google academic search engine, using various relevant keywords. Furthermore, some highly persuasive references in the obtained literature were also tracked.

### 5.2. Analyses of ARG Generation, Spread and Elimination from the OE-OH Concept

From the OE-OH concept, the generation, spread, and elimination of ARGs along with different time nodes of the Earth’s evolution were analyzed and sorted out from various aspects of abiotic and biotic factors [33,34,42], using reasoning, deductive, and inductive methods based on the self-regulation of ecosystems [13,14,15], together with the supporting literature unsystematically searched from PubMed database and Google academic search engine using some relevant search terms. These especially include the emergence and distribution of ARGs emerging before humans, ARG regulation by ecosystems such as the generation, spread, weakening, and elimination of ARGs before the industrial production and use of antibiotics, and the sharp increase in ARGs after the industrial production and use of antibiotics, which together impact the self-regulation and self-balance of ecosystems for ARGs.

### 5.3. Measures Combating AMR from the OE-OH Concept Based on Ecosystems

Regarding the use of antibiotics as an intervention on ecosystems, some important measures for the research and development, application, and management of antibiotics are suggested, from the OE-OH prospective, for maintaining the balance of ecosystems regulating ARGs, learning from the approach of problem management [121,122].

## 6. Future Directions

Drawing on the OE-OH concept, the following research directions are proposed for future exploration: (1) referring to Lamarck’s theory and niche construction, emphasizing the interactions among various factors in host ecosystems and strengthening research on the proactive defense mechanisms of microbe, for developing new antibiotics with strong selectivity that cause minimal disruption to human ecosystems; (2) conducting in-depth investigations into ARGs in problem-oriented samples from both designed terrestrial and marine environments, which can not only can help to elucidate the dissemination of ARGs but can also provide additional evidence for understanding the Earth’s evolution, plate tectonics, and human migration; (3) emphasizing the crucial role of the self-regulation of ecosystems on addressing AMR, strengthening research on the regulation of plants on the ecosystem of human bodies to prevent microbial infection, and encouraging the research and development of antibiotics in combination with plant-derived antimicrobial ingredients; (4) increasing efforts to study the dilution and degradation of antibiotics in ecosystems, as well as the weakening and elimination of ARGs, and developing methods and technologies that simulate the elimination of antibiotics and ARGs within ecosystems; and (5) based on the self-regulatory capacity of ecosystems in the elimination of antibiotics and ARGs, conducting research on the schemes of population distribution and urban settlements for combatting AMR.

## 7. Conclusions

The OE-OH concept has been clarified and refined, which includes (1) a dual mutation pattern of primarily proactive evolution aligning with Lamarck’s theory and occasionally passive selection in accordance with Darwinian evolution theory for microbial resistance mutation, adhering to the principle of parsimony; (2) the theoretical logic of this concept, which is based on the robust self-regulatory capacity and perpetual reconstruction of ecosystems, and a fundamental mathematical model for the renewal of ARGs within the ecosystem; and (3) the notion that the farther the evolutionary distance between species, the weaker the antagonistic effect of the secondary metabolites they produce on each other, and the more difficult it is for them to develop resistance to each other. Derived from this concept, it deduced that AMR and ARGs emerged 3.5 billion years ago and existed in every corner of the Earth prior to the use of antibiotics by humans. Regarded as an intervention on ecosystems, the use of antibiotics, particularly their overuse and abuse, has posed a concern that transcends the ecosystem’s self-regulatory capacity. Based on these, several crucial measures derived from the OE-OH concept are proposed for combatting AMR. These measures place a strong emphasis on simulating and leveraging the self-regulatory mechanisms of ecosystems, advocating for the minimization of antibiotic use and emissions, preventing the excessive density and size of urban populations, and encouraging antibiotics used in combination with plant-derived antimicrobial ingredients. Finally, it is essential to implement the OH Joint Plan of Action from the OE-OH perspective, emphasizing the key role of utilizing the self-regulation of ecosystems in addressing AMR.

## Figures and Tables

**Figure 1 antibiotics-14-00920-f001:**
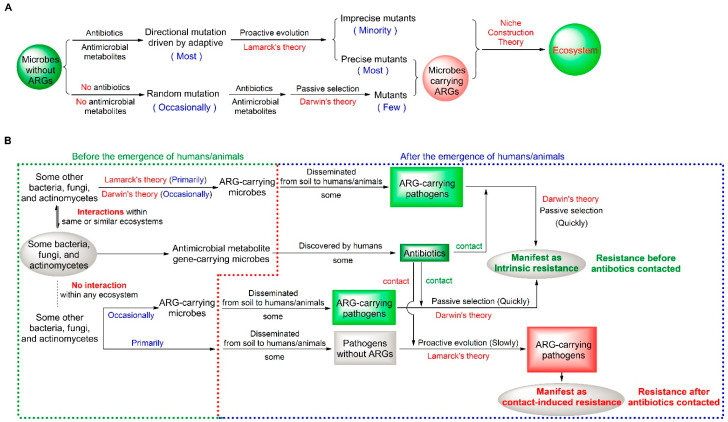
A dual mutation pattern of microbial resistance (**A**) and the evolutionary origins of current antimicrobial resistance (AMR) (**B**). ARGs, antibiotic resistance genes.

**Figure 2 antibiotics-14-00920-f002:**
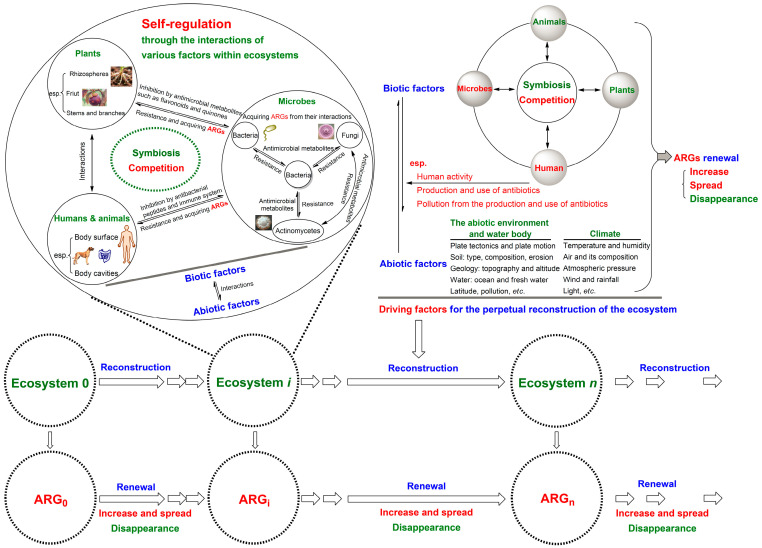
Schematic for the theoretical foundation of the One Earth–One Health concept. This includes the theoretical logic and driving factors of ARG renewal within an ecosystem based on the robust self-regulatory capacity and perpetual reconstruction of ecosystems; the ecosystem can be the entire Earth’s ecosystem or its various sub-ecosystems; ecosystems 0, i, and n indicate the ecosystem at different time points during its evolution, and their antibiotic resistance genes (ARGs) correspondingly renew with the perpetual reconstruction of the ecosystem.

**Figure 3 antibiotics-14-00920-f003:**
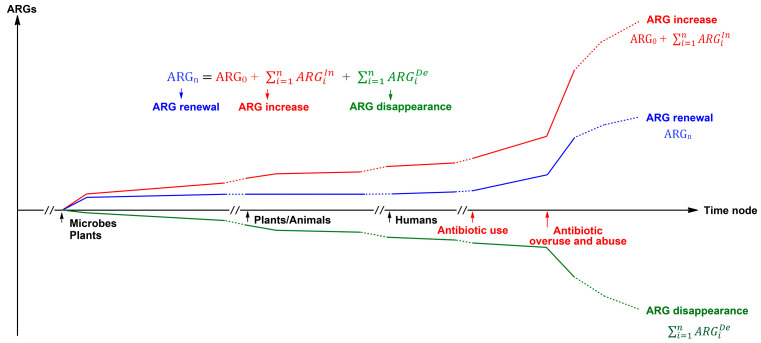
A schematic for the amount of antibiotic resistance genes (ARGs, y) within an ecosystem changing with typical evolutionary time nodes (Time node, x), according to a fundamental mathematical model. The ecosystem can be the entire Earth’s ecosystem or its various sub-ecosystems, and the typical time nodes include the emergence of microbes, plants, animals, and humans and the use of antibiotics by humans.

**Figure 4 antibiotics-14-00920-f004:**
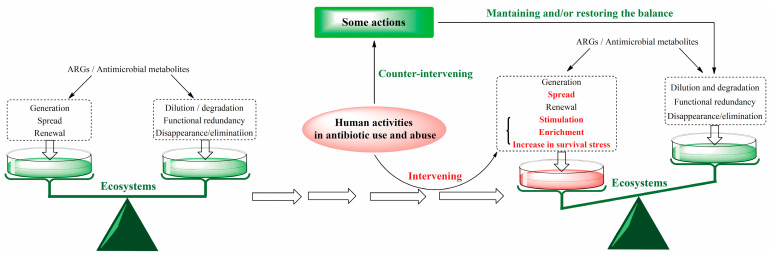
Analysis schematic of some imperative actions taken to offset the imbalance risk of ecosystems caused by the antibiotic use exceeding the self-regulation and balancing capacity of ecosystems, for combating the AMR from the OE-OH concept. ARGs, antibiotic resistance genes.

## Data Availability

No data were generated for this manuscript.

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
