# Peer review of "One Earth–One Health to Combat Antimicrobial Resistance Having a Dual Mutation Pattern, Based on the Robust Self-Regulation and Perpetual Reconstruction of Ecosystems"

_antibiotics, 2025, doi:10.3390/antibiotics14090920_

Round 1

Reviewer 1 Report

Comments and Suggestions for Authors

The manuscript with antibiotic ID number 3837049 is written in a highly scientific language and the presentation point of view is very interesting.

In my opinion, there are only a few points for correction:

On line 21, “AGR” occurs for the first time, but only as an abbreviation. On line 121, it occurs as a full explanation. Try to correct it on line 21.
There were too many keywords - sixteen. The author's instructions in the journal “Antibiotics” state that keywords should be from 3 to 10. In my opinion, it is necessary to add as a new keyword also “ABR” or “antibiotic resistance”.
In section 4.4. there were no references, and in section 5.2. there were also no references and it was written “PubMed database and Google academic search engine”, which was not sufficient evidence. In my opinion, references need to be added.
REFERENCES - 78% are from the last 5 years.

Author Response

Dear Reviewer,

My co-authors and I are very grateful to you for your careful review, good comments, kind reminder, valuable suggestions, and great help to improve our works. We have amended the manuscript according to the issues raised by you, and have pleasure to submit the revised version, together with the response to all points, for your consideration.

Many thanks for your kind attention!

Yours sincerely,

Ganjun Yuan

Here are our answers to your comments.

Comments and Suggestions for Authors

The manuscript with antibiotic ID number 3837049 is written in a highly scientific language and the presentation point of view is very interesting.

Response: Thank you very much for your careful review, and good comments!

In my opinion, there are only a few points for correction:

1. On line 21, “AGR” occurs for the first time, but only as an abbreviation. On line 121, it occurs as a full explanation. Try to correct it on line 21.

Response 1: Thank you very much for your careful review, and valuable suggestion! We have corrected it on Lines 21 to 22 in the revised manuscript.

2. There were too many keywords - sixteen. The author's instructions in the journal “Antibiotics” state that keywords should be from 3 to 10. In my opinion, it is necessary to add as a new keyword also “ABR” or “antibiotic resistance”.

Response 2: Thank you very much for your careful review, kind reminder, and valuable suggestion to improve our work! Adding a keyword “antibiotic resistance” is very consistent with the actual content of our manuscript. Therefore, we have inserted “antibiotic resistance” as a new keyword according to your suggestion. Another, we have deleted some keywords (such as mutation, ecosystem, antibiotic, climate, and parsimony) to fit the requirement of the journal “Antibiotics” state that keywords should be less than or equal to 10, for your consideration.

3. In section 4.4. there were no references, and in section 5.2. there were also no references and it was written “PubMed database and Google academic search engine”, which was not sufficient evidence. In my opinion, references need to be added.

REFERENCES - 78% are from the last 5 years.

Response 3: Thank you very much for your careful review, kind reminder, and valuable suggestion to improve our work!

For section 4.4., we have made many revisions, and cited three new references 83 to 85, and refence 18 in the revised manuscript to support our viewpoint, for your consideration!

For section 5.2., we have inserted references 33, 34, and 42 in the revised manuscript, and provided more details for achieving the analyses of the generation, spread and elimination of ARGs. Moreover, we have inserted references 13 to 15 for that readers can find the interpretation of the self-regulation of ecosystems, for your consideration!

Other revisions

Besides above revisions mentioned, we had carefully performed extensive revision throughout the manuscript including references, such as the spelling, formatting, syntax, linguistic edit and expression, especially for the introduction and section 4, for your consideration.

Thank you very much for your careful review, good comments, valuable suggestion, and great help to improve our work!

Reviewer 2 Report

Comments and Suggestions for Authors

This paper provides a clear overview of the One Earth–One Health concept, highlighting its relevance in designing policies to combat the global crisis of antimicrobial resistance (AMR). The authors have discussed the dual mutation patterns underlying bacterial resistance, as well as the evolutionary origins of antibiotic resistance, and have outlined several distinctive strategies to address this issue.  Overall, the paper addresses a timely and significant topic. Addressing the below points will strengthen the manuscript and enhance its clarity for readers.

  1. Please explain the term One Health Joint Plan of Action (2022–2026) in detail, including the international agencies involved in this initiative (Line 44).
  2. In the introduction, I suggest first presenting the general concept of One Earth–One Health, followed by its correlation with AMR or ARGs.
  3. There is unnecessary repetition of statements such as “microbes emerged about 3.5 billion years ago.” Please revise to avoid redundancy.
  4. Recheck reference (36) to confirm whether it adequately supports the statement: “Likely, the genetic information of microorganisms producing these natural products has existed on the Earth before the emergence of humans.”
  5. Provide an appropriate reference for Lines 227–228. Additionally, please cite the source of data used to generate Figure 3.
  6. Review the alignment and formatting of the text in Box 1 (Factors for the weakening and elimination of ARGs).
  7. The emergence of antimicrobial resistance in microbes against novel synthetic peptides, which possess one or multiple targets within microbes, has already been reported. How do the authors view this phenomenon through the lens of the One Earth–One Health (OE–OH) framework?
  8. In Section 4.1, the discussion on the role of plants in regulating or controlling antimicrobial-resistant microbes is insufficient. Please elaborate further.
  9. Some suggestions appear highly hypothetical and difficult to achieve in the current scientific and ethical landscape (e.g., Section 4.3). These should be revised or better contextualized.
  10. There are contradictory statements, for example between Lines 288–291 and Section 4.4. Please revise to maintain consistency.

Author Response

Dear Reviewer,

My co-authors and I are very grateful to you for your careful review, good comments, valuable suggestions, and great help to improve our work. We have carefully checked the manuscript again, and have pleasure to submit the revised version, together with the response to your comments, for your consideration.

Many thanks for your kind attention!

Yours sincerely,

Ganjun Yuan

Here are our answers to your comments.

Comments and Suggestions for Authors

This paper provides a clear overview of the One Earth–One Health concept, highlighting its relevance in designing policies to combat the global crisis of antimicrobial resistance (AMR). The authors have discussed the dual mutation patterns underlying bacterial resistance, as well as the evolutionary origins of antibiotic resistance, and have outlined several distinctive strategies to address this issue.  Overall, the paper addresses a timely and significant topic. Addressing the below points will strengthen the manuscript and enhance its clarity for readers.

Response: Thank you very much for your careful review, good comments, valuable suggestions, and great help to improve our work!

1. Please explain the term One Health Joint Plan of Action (2022–2026) in detail, including the international agencies involved in this initiative (Line 44).

Response 1: Thank you very much for your careful review, valuable suggestions, and kind reminder! We have provided the international agencies in this initiative (Lines 45 to 47 in the revised manuscript) and the relevant information (Lines 49 to 51) to the One Health Joint Plan of Action (2022–2026), for your consideration.

2. In the introduction, I suggest first presenting the general concept of One Earth–One Health, followed by its correlation with AMR or ARGs.

Response 2: Thank you very much for your valuable suggestions, and great help to improve our work! According to your suggestion, it will greatly enhance its clarity for readers and enable readers to quickly obtain the information they most desire from this manuscript. The application of OE–OH concept is not only used to address AMR, but also other health threats to humans, animals, plants, and environment, even other public problems. Considering that it is the aim of this manuscript that presents the application of OE-OH concept in addressing AMR, we first present the crisis of AMR and current efforts to address AMR. Inspired by your suggestion, we inserted a paragraph (Paragraph 3) in the introduction for providing a brief introduction to the OE-OH concept for your consideration, and will first present the general concept of OE-OH in following manuscript which present the applications of OE-OH concept in diverse aspects. Thank you very much again for your valuable suggestion and great help to improve our current and following work!

3. There is unnecessary repetition of statements such as “microbes emerged about 3.5 billion years ago.” Please revise to avoid redundancy.

Response 3: Thank you very much for your careful review, and kind reminder! We checked the manuscript, and revised the repetition of statements for your consideration. For example, in section 3.1, we have revised “From the evolutionary history of the Earth, it is known that microbes emerge about 3.5 billion years ago” as “As noted earlier, microbes emerge 3.5 billion years ago” (Line 228 in the revised manuscript), and revised “Since antimicrobial metabolites generated before 3.5 billion years ago” as “Since antimicrobial metabolites generated with the formation of microbial ecosystems” (Line 247 in the revised manuscript).

4. Recheck reference (36) to confirm whether it adequately supports the statement: “Likely, the genetic information of microorganisms producing these natural products has existed on the Earth before the emergence of humans.”

Response 4: Thank you very much for your careful review, and kind reminder! We have checked it, and updated the reference by three new references 36 to 38 (in the revised manuscript) to adequately supports the statement, for your consideration.

5. Provide an appropriate reference for Lines 227–228. Additionally, please cite the source of data used to generate Figure 3.

Response 5: Thank you very much for your careful review, and valuable suggestions! We have provided an appropriate reference (Ref. 41) for Lines 227–228 (Lines 245-246 in the revised manuscript). Additionally, Figure 3 is a schematic diagram and first presented, and which is schematically generated from the fundamental model according to the time nodes of the Earth’s evolution history. Therefore, no source of data can be cited. Since Figure 3 is first cited at Line 144 (in the original manuscript), it has been deleted in the revised manuscript to avoid possible misunderstanding.

6. Review the alignment and formatting of the text in Box 1 (Factors for the weakening and elimination of ARGs).

Response 6: Thank you very much for your careful review, and kind reminder! We have made corresponding adjustments for the alignment and formatting of the text in Box 1 in the revised manuscript, for your consideration.

7. The emergence of antimicrobial resistance in microbes against novel synthetic peptides, which possess one or multiple targets within microbes, has already been reported. How do the authors view this phenomenon through the lens of the One Earth–One Health (OE–OH) framework?

Response 7: It is a good topic worthy of exploration. There is a universal expectation for the development of antibiotics that do not trigger microbial resistance. However, from the OE-OH perspective, it is unrealistic for humans to discover antibiotics which microorganisms will never develop resistance. This also holds true for multi-target antibiotics whether they are natural, fully synthetic products (such as sulfonamides and quinolones), or antimicrobial peptides (AMPs). The only difference lies in the timeframe required to detect the emergence of resistant bacterial strains. This also represents one of the differences between OE–OH and OH. Nevertheless, to combat AMR, it remains necessary to develop new antibiotics that are difficult to trigger resistance as much as possible, and a sufficient reserve of these new antibiotics should be ensured.

Natural AMPs are endogenously present in both animals (e.g., frogs) and the human body (some AMPs formed by protein degradation). From OE-OH perspective, these peptides are evolutionary products of animals and humans competing for ecological niches within ecosystems. Bacteria will also develop resistance to AMPs after exposure, with this process being dependent on the temporal and spatial context of contact. The pathways and timeline of resistance development can be analyzed by referencing Figure 1B in this manuscript.

For synthetically designed AMPs including multi-target ones, if structural analogs of these peptides already exist in the Earth’s environment, resistant bacteria may be detected even before the synthetic AMP is officially launched. If the synthetic AMP has a structure entirely distinct from that of naturally occurring AMPs in the Earth’s environment, bacterial resistance will still emerge, perhaps, shortly after exposure.

The above represents my perspectives from the OE-OH for your consideration. Thank you very much for your good topic.

8. In Section 4.1, the discussion on the role of plants in regulating or controlling antimicrobial-resistant microbes is insufficient. Please elaborate further.

Response 8: Thank you very much for your careful review, kind reminder, and valuable suggestion! We have inserted a sentence as “This can be achieved through their regulatory effects on the body’s inflammatory and immune responses, their direct antibacterial, antifungal, and antiviral actions, as well as their ability to enhance the inhibitory activities of metabolites produced by probiotics against antimicrobial-resistant microbes.” for further elaborating the regulatory effects of plants on the body’s ecosystems in the balance between the bodies and microbes.

9. Some suggestions appear highly hypothetical and difficult to achieve in the current scientific and ethical landscape (e.g., Section 4.3). These should be revised or better contextualized.

Response 9: Thank you very much for your careful review, kind reminder, and valuable suggestion! We have made many revisions for this section including its title, providing more accurate expression. Moreover, we have inserted a new citation (Ref. 82) to support corresponding viewpoints for your consideration.

10. There are contradictory statements, for example between Lines 288–291 and Section 4.4. Please revise to maintain consistency.

Response 10: Thank you very much for your careful review, and kind reminder! We have made comprehensive revisions to Section 4.4 for maintaining consistency, for your consideration.

Other revisions

Besides above revisions mentioned, we had carefully performed extensive revision throughout the manuscript including references, such as the spelling, formatting, syntax, linguistic edit and expression, especially for the introduction and section 4, for your consideration.

Thank you very much for your careful review, good comments, valuable suggestion, and great help to improve our work! 

Reviewer 3 Report

Comments and Suggestions for Authors

The manuscript antibiotics-3837049, structured as a communication, has a challenging purpose: to explain the innovative concept "One Earth-One Health" and display its usefulness in antimicrobial resistance consideration. The manuscript is based on 127 references, most of them very recently published. 

The communication's design is complex, and all data are well explained; relevant figures support all data presentation.

The strengths of this communication are significant: 

1. The OE-OH concept has a multivalent presentation:

  • revealing a dual mutation pattern of bacterial resistance;
  • offering a theoretical logic approach based on ecosystems;
  • building a fundamental mathematical model for the ARGs renewing with the ecosystem.

2. The ARGs are deeply analyzed through the OE-OA concept, and the Impact of antibiotic use on the self-regulation of ecosystems to ARGs is revealed.

3. A series of AMR prevention measures is displayed, with supporting details.  

Some minor suggestions are available:

It would be better to change ABR to a more suitable AMR for better understanding. ABR is quite unusual; the first explanation of this abbreviation significantly differs from this. Moreover, the authors refer to all pathogens, not only to bacteria.  

The authors are encouraged to show the basis of the concept name (OE-OH) because the reviewer found a similar name in another field. 

Author Response

Dear Reviewer,

My co-authors and I are very grateful to you for your careful review, good comments, kind reminder, valuable suggestions, and great help to improve our work. We have amended the manuscript according to the issues raised by you, and have pleasure to submit the revised version, together with the response to your comments, for your consideration.

Many thanks for your kind attention!

Yours sincerely,

Ganjun Yuan

Here are our answers to your comments.

Comments and Suggestions for Authors

The manuscript antibiotics-3837049, structured as a communication, has a challenging purpose: to explain the innovative concept "One Earth-One Health" and display its usefulness in antimicrobial resistance consideration. The manuscript is based on 127 references, most of them very recently published.

The communication's design is complex, and all data are well explained; relevant figures support all data presentation.

The strengths of this communication are significant:

  1. The OE-OH concept has a multivalent presentation: 1) revealing a dual mutation pattern of bacterial resistance; 2) offering a theoretical logic approach based on ecosystems; 3) building a fundamental mathematical model for the ARGs renewing with the ecosystem.
  2. The ARGs are deeply analyzed through the OE-OA concept, and the impact of antibiotic use on the self-regulation of ecosystems to ARGs is revealed.
  3. A series of AMR prevention measures is displayed, with supporting details.

Response: Thank you very much for your careful review, good comments, and insightful summaries to our manuscript!

Some minor suggestions are available:

1. It would be better to change ABR to a more suitable AMR for better understanding. ABR is quite unusual; the first explanation of this abbreviation significantly differs from this. Moreover, the authors refer to all pathogens, not only to bacteria.

Response 1: You are right! It would be better to change ABR to a more suitable AMR. Therefore, we have made the revisions. Reminded by you, we have carefully checked the manuscript and revised relevant expressions throughout the manuscript. For example, the expression “bacterial resistance” and “bacteria” were revised as “antimicrobial resistance” or “microbial resistance” and “microbes”, respectively. Thank you very much for your careful review, valuable suggestions, and great help to our work!

2. The authors are encouraged to show the basis of the concept name (OE-OH) because the reviewer found a similar name in another field.

Response 2: Thank you very much for your careful review, kind reminder, and valuable suggestion! We have inserted a paragraph to define the OE-OH in section introduction for distinguishing from similar name, and cited our previous works published as the basis of the OE-OH. Another, we also have presented the difference between OE-OH and OH.

Other revisions

Besides above revisions mentioned, we had carefully performed extensive revision throughout the manuscript including references, such as the spelling, formatting, syntax, linguistic edit and expression, especially for the introduction and section 4, for your consideration.

Thank you very much for your careful review, good comments, valuable suggestion, and great help to improve our work!